

# Renormalizable extension of the Abelian Higgs-Kibble model with a dim.6 derivative operator

Daniele Binosi[1] and Andrea Quadri[2*]

**1** European Centre for Theoretical Studies in Nuclear Physics and Related Areas (ECT*)
and Fondazione Bruno Kessler, Villa Tambosi, Strada delle Tabarelle 286,
I-38123 Villazzano (TN), Italy
**2** INFN, Sez. di Milano, via Celoria 16, I-20133 Milan, Italy

⋆ andrea.quadri@mi.infn.it

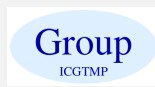

## Abstract

We present a new approach to the consistent subtraction of a non power-counting renormalizable extension of the Abelian Higgs-Kibble (HK) model supplemented by a dim. 6 derivative-dependent operator controlled by the parameter $z$. A field-theoretic representation of the physical Higgs scalar by a gauge-invariant variable is used in order to formulate the theory by exploiting a novel differential equation, controlling the dependence of the quantized theory on $z$. These results pave the way to the consistent subtraction by a finite number of physical parameters of some non-power-counting renormalizable models possibly of direct relevance to the study of the Higgs potential at the LHC.

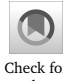
# 1 Introduction

In the quest for new physics at the LHC a significant role has been recently played on the theoretical side by the Standard Model (SM) Effective Field Theories [1–3]. Deviations from the SM Lagrangian are described by a set of gauge-invariant higher-dimensional operators suppressed by some large energy scale $\Lambda$. The resulting theory is no more power-counting renormalizable and therefore more and more ultraviolet (UV) divergences arise as more loops are included. Their subtraction requires the introduction of more and more higher-dimensional operators compatible with the symmetries of the model.

Power-counting renormalizable theories on the other hand are defined in terms of a finite number of physical parameters in one-to-one correspondence with the finite number of operators required to subtract the UV divergences of the one-particle irreducible (1-PI) amplitudes to all orders in the loop expansion (once linear wave-function renormalization has been taken into account).

This is at variance with the increasing number of higher-dimensional operators required to make effective field theories finite as higher perturbative orders are included. Consequently, effective field theories preserve predictivity only up to the energy scale $\Lambda$: below $\Lambda$ only a finite number of higher dimensional operators are physically relevant and in this sense a finite number of physical parameters control physical observables (up to the relevant energy scale).

The question of the minimal set of independent physical operators required to renormalize an effective field theory is a subtle question. First of all one must take into account redundacies associated with the equations of motion [4], or equivalently by generalized field redefinitions that are in general non-polynomial and prove essential in order to consistently subtract UV divergences by local counter-terms [5].

Moreover, it has been recently advocated [4,6] that additional relations between seemingly independent UV divergent amplitudes are easier to derive within a particular choice of gauge-invariant field coordinates [7–9].

For instance, in the usual formalism of the Abelian Higgs-Kibble model, the complex scalar field $\phi = \frac{1}{\sqrt{2}}(v + \sigma + i\chi)$ is used, $v$ being the vacuum expectation value of $\phi$, $\sigma$ the field describing the physical scalar mode and $\chi$ the pseudo-Goldstone field. $\phi$ transforms in the fundamental representation of the gauge group U(1), $\delta\phi = ie\alpha\phi$ with $\alpha$ the infinitesimal gauge transformation and $e$ the U(1) gauge coupling constant.

One might also consider the alternative choice of using the gauge invariant combination

$$\phi^\dagger \phi - \frac{v^2}{2} \sim v X_2 \,,$$

in order to represent the physical scalar mode (this is the so-called $X$-formalism, based on the set of auxiliary fields $X_2$ and the Lagrange multiplier $X_1$).

The resulting theory has been described at length in [4,6,10–12] and its tree-level vertex functional is reported in Eq.(A.1). It is physically equivalent to the Abelian Higgs-Kibble model, after going on-shell with both $X_1$ and $X_2$.

At variance with the ordinary formalism, the set of functional identities of the theory in the $X$-formalism is richer. For instance, 1-PI amplitudes involving at least one $X_1$ or $X_2$-fields are uniquely fixed by the $X_{1,2}$-functional equations in Eqs.(B.3,B.4) in terms of amplitudes without.

More importantly, it turns out that the $X_2$-equation of the Abelian Higgs-Kibble model admits a unique deformation, compatible with all the symmetries of the theory and associated with the addition to the classical action of a bilinear operator in $X_2$, see the first term in the second line of Eq.(A.1). At $z = 0$ we recover the power-counting renormalizable Abelian Higgs-Kibble model, while at $z \neq 0$ we obtain a non power-counting renormalizabile theory

physically equivalent to the one generated by the introduction of the dim.6 operator

$$\frac{z}{2}\partial^\mu X_2 \partial_\mu X_2 \sim \frac{z}{2v^2}\partial^\mu(\phi^\dagger\phi)\partial_\mu(\phi^\dagger\phi). \tag{1}$$

A crucial remark is that in the $X$-formalism the parameter $z$ enters classically only in the quadratic part of the classical action, while in the standard approach it also appears in the interaction vertices. This property allows one to derive in the $X$-formalism an extremely powerful differential equation.

By solving the latter equation, one obtains a unique prescription for the amplitudes of the non-power-counting renormalizable model at $z \neq 0$ in terms of those at $z = 0$.

If the theory at $z = 0$ is power-counting renormalizable (as in the case we deal with in the present paper, for the sake of definiteness), the model at $z \neq 0$ is defined in terms of the same (finite, to all orders in perturbation theory) number of physical parameters plus $z$.

If, on the other hand, the model at $z = 0$ is an effective field theory, the results of the present paper show that the addition of the dim.6 interaction in Eq.(1) comes at no cost, since the complete dependence of the amplitudes at $z \neq 0$ is still uniquely determined algebraically by the $z$-differential equation in terms of the amplitudes at $z = 0$.

The relevant parameters up to the scale energy $\Lambda$ are those of the effective theory at $z = 0$ plus $z$. This is a highly non-trivial result that follows from the $z$-differential equation.

## 2 The z-differential equation

The starting point is the diagonalization of the quadratic part in the scalar sector, that can be achieved by the field redefintion

$$\sigma = \sigma' + X_1 + X_2. \tag{2}$$

The propagators read

$$\Delta_{\sigma'\sigma'} = \frac{i}{p^2 - m^2}, \qquad \Delta_{X_1 X_1} = -\frac{i}{p^2 - m^2}, \qquad \Delta_{X_2 X_2} = \frac{i}{(1+z)p^2 - M^2}. \tag{3}$$

In this basis the dependence on the parameter $z$ only arises via the $X_2$-propagator. Introducing then the differential operator

$$\mathcal{D}_z^{M^2} = (1+z)\partial_z + M^2 \partial_{M^2}, \tag{4}$$

one finds that $\Delta_{X_2 X_2}$ is an eigenvector of $\mathcal{D}_z^{M^2}$ with eigenvalue -1:

$$\mathcal{D}_z^{M^2}\Delta_{X_2 X_2}(k^2, M^2) = -\Delta_{X_2 X_2}(k^2, M^2). \tag{5}$$

The argument generalizes to diagrams with a given number of internal $X_2$-lines. Let us collectively denote with $\Phi$ the set of fields and external sources of the theory, and let us indicate with $p_i$ (with $i = 1, \ldots, r$) their external momenta, with $\Phi_i = \Phi(p_i)$ and $p_r = -\sum_1^{r-1} p_i$; in this way a $n$-loop 1-PI Green's function $\Gamma^{(n)}_{\Phi_1 \cdots \Phi_r}$ with $r$ $\Phi_i$ insertions can be decomposed as the sum of all 1-PI diagrams with external legs $\Phi_1 \cdots \Phi_r$ with zero, one, two,..., $\ell$ internal $X_2$-propagators, i.e.,

$$\Gamma^{(n)}_{\Phi_1 \cdots \Phi_r} = \sum_{\ell \geq 0} \Gamma^{(n;\ell)}_{\Phi_1 \cdots \Phi_r}. \tag{6}$$

Then by applying the differential operator $\mathcal{D}_z^{M^2}$ we find

$$\mathcal{D}_z^{M^2}\Gamma^{(n;\ell)}_{\Phi_1 \cdots \Phi_r} = -\ell\,\Gamma^{(n;\ell)}_{\Phi_1 \cdots \Phi_r} \implies \mathcal{D}_z^{M^2}\Gamma^{(n)}_{\Phi_1 \cdots \Phi_r} = -\sum_{\ell \geq 0} \ell\,\Gamma^{(n;\ell)}_{\Phi_1 \cdots \Phi_r}. \tag{7}$$

Hence we see that the subdiagrams with a fixed number $\ell$ of internal $X_2$-lines are eigenvectors of the $\mathcal{D}_z^{M^2}$ with eigenvalue $\ell$. The most general solution to this equation (of the homogeneous Euler's type) reads (indicating explicitly only the dependence on the parameters $z$ and $M^2$)

$$\Gamma_{\Phi_1\cdots\Phi_r}^{(n;\ell)}(z, M^2) = \frac{1}{(1+z)^\ell}\Gamma_{\Phi_1\cdots\Phi_r}^{(n;\ell)}(0, M^2/(1+z)). \tag{8}$$

Thus, amplitudes at $z \neq 0$ in each $\ell$-sector are obtained from those at $z = 0$ by dividing them by the $(1+z)^\ell$ factor and rescaling by $(1+z)$ the square of the Higgs mass $M^2$.

Otherwise said, in the $X$-formalism the existence of the $z$-differential equation implies that the deformed theory at $z \neq 0$ can be fully characterized once one knows the boundary conditions given by the amplitudes of the power-counting renormalizable theory at $z = 0$.

## 2.1 ST identities in the $\ell$-sector

Another crucial property of the $X$-formalism is that the ST identities separately hold true in each $\ell$-sector. The proof of this statement can be found in [12] and relies on the gauge-invariance of the $X_2$-field.

At order $n$ in the loop expansion we get a set of ST identities, one for each $\ell$:

$$\mathcal{S}_0\left(\Gamma^{(n;\ell)}\right) + \sum_{j=1}^{n-1}\sum_{i=0}^{\ell}\left(\Gamma^{(j;i)}, \Gamma^{(n-j;\ell-i)}\right) = 0. \tag{9}$$

Such identities encode the conditions required to guarantee physical unitarity of the theory (i.e., the cancellation of the intermediate ghost states). Since $X_2$ is gauge-invariant, it is physically sensible that it does not participate to such cancellations and therefore that the quartet mechanism [13–15] is at work separately for each sector with a given number $\ell$ of internal $X_2$-lines.

### 2.1.1 Normalization conditions

The normalization conditions that must be imposed in the theory at $z = 0$ can also be consistently decomposed according to the degree induced by the number of internal $X_2$-lines.

For instance, the on-mass shell normalization condition for the vector meson is obtained by requiring that the position of the pole of the physical components of the vector meson does not shift with respect to the one at tree level and that the residue of the propagator on the pole is one, i.e.

$$\text{Re } \Sigma_T(M_A^2) = 0, \qquad \text{Re } \left.\frac{\partial \Sigma_T(p^2)}{\partial p^2}\right|_{p^2=M_A^2} = 0. \tag{10}$$

In the above equation we have denoted by $\Sigma_T$ the transverse component of the two-point 1-PI gauge function:

$$\Gamma_{A^\mu A^\nu} = g_{\mu\nu}(p^2 - M_A^2) + \left(g_{\mu\nu} - \frac{p_\mu p_\nu}{p^2}\right)\Sigma_T(p^2) + \frac{p^\mu p^\nu}{p^2}\Sigma_L(p^2). \tag{11}$$

These conditions can be matched by finite renormalizations involving the following ST (and gauge-) invariant operators (we use the notation of Ref. [6]):

$$\lambda_4 \int \mathrm{d}^4x\, (D^\mu\phi)^\dagger D_\mu\phi \supset \frac{\lambda_4 v}{2}\int \mathrm{d}^4x\, A_\mu^2, \qquad \frac{\lambda_8}{2}\int \mathrm{d}^4x\, F_{\mu\nu}^2 \supset \lambda_8\int \mathrm{d}^4x\, A_\mu(\Box g^{\mu\nu} - \partial^\mu\partial^\nu)A_\nu. \tag{12}$$

Now, since the ST identities hold true separately at each $\ell$-order, we can project the normalization condition Eq.(10) at the relevant $\ell$-order and at order $n$ in the loop expansion:

$$\text{Re } \Sigma_T^{(1;\ell)}(M_A^2) + \nu \lambda_4^{(1;\ell)} = 0, \qquad \text{Re } \left.\frac{\partial \Sigma_T^{(1;\ell)}}{\partial p^2}\right|_{p^2 = M_A^2} - 2M_A^2 \lambda_8^{(1;\ell)} = 0. \tag{13}$$

As can be seen from the above equation, on mass shell renormalization conditions respect the layers in $\ell$ and consequently the $z$-differential equation.

Otherwise said, once the appropriate normalization conditions are enforced at order $n$ in the loop expansion at $z = 0$, Eq.(8) fixes the 1-PI amplitudes of the theory at $z \neq 0$ in a unique way.

## 3 Conclusion

We have obtained a differential equation that controls the deformation of the Abelian Higgs-Kibble model induced by the dim.6 operator

$$\frac{z}{2}\partial^\mu X_2 \partial_\mu X_2 \sim \frac{z}{2}\partial^\mu(\phi^\dagger\phi)\partial_\mu(\phi^\dagger\phi).$$

The solution to the differential equation is uniquely defined in terms of the boundary conditions of the (renormalized) amplitudes of the theory at $z = 0$. This allows one to define the corresponding non-power-counting renormalizable theory in a way that it only depends on the same number of physical parameters of the model at $z = 0$ (either the finite ones, to all orders in perturbation theory, if the model at $z = 0$ is power-counting renormalizable, or those relevant up to the energy scale $\Lambda$, if the model at $z = 0$ is an effective field theory), and $z$.

The results obtained so far for the Abelian gauge group can be generalized to the full electroweak SU(2)×U(1) theory. This is of particular interest, since one could obtain an extension of the SM and of Beyond-the-Standard-Model (BSM) theories by a derivative-dependent dim.6 operator, that still can be defined at the quantum level in a consistent way (to all orders in $z$). Within this framework, applications to phenomenology should also be studied. In particular one could study the BSM corrections to the SM Higgs potential, that are expected to be explored at the LHC experimental program.

Another interesting problem is whether the present construction can be extended to gauge-invariant fields representing the gauge and fermion degrees of freedom. We hope to report on these issues soon.

## Acknowledgements

**Funding information**   A partial financial support by INFN is acknowledged.

# A  Classical vertex functional in the X-formalism

The classical vertex functional is given by:

$$
\Gamma^{(0)} = \int \mathrm{d}^4 x \left[ -\frac{1}{4} F^{\mu\nu} F_{\mu\nu} + (D^\mu \phi)^\dagger (D_\mu \phi) - \frac{M^2 - m^2}{2} X_2^2 - \frac{m^2}{2v^2} \left( \phi^\dagger \phi - \frac{v^2}{2} \right)^2 \right.
$$
$$
+ \frac{z}{2} \partial^\mu X_2 \partial_\mu X_2 - \bar{c}(\Box + m^2)c + \frac{1}{v}(X_1 + X_2)(\Box + m^2)\left( \phi^\dagger \phi - \frac{v^2}{2} - vX_2 \right)
$$
$$
+ \frac{\xi b^2}{2} - b(\partial A + \xi e v \chi) + \bar{\omega}\left( \Box \omega + \xi e^2 v(\sigma + v)\omega \right)
$$
$$
\left. + \bar{c}^* \left( \phi^\dagger \phi - \frac{v^2}{2} - vX_2 \right) + \sigma^*(-e\omega\chi) + \chi^* e\omega(\sigma + v) \right]. \tag{A.1}
$$

In the above equation $D_\mu$ is the covariant derivative

$$
D_\mu = \partial_\mu - ieA_\mu. \tag{A.2}
$$

The first line of Eq.(A.1) is the classical action of the Abelian Higgs-Kibble model. By going on-shell with $X_1$ and imposing the constraint

$$
X_2 = \frac{1}{v} \left( \phi^\dagger \phi - \frac{v^2}{2} \right), \tag{A.3}
$$

we recover the usual quartic Higgs potential with coupling $\sim -\frac{M^2}{2v^2}$. Indeed one can prove [6] that the only physical parameter is $M$, $m$ cancelling out in physical quantities. The first term of the second line contains the deformation proportional to the parameter $z$. By going on-shell with $X_1$ we obtain the dimension-six derivative operator $\sim \frac{z}{2v^2} \partial^\mu (\phi^\dagger \phi) \partial_\mu (\phi^\dagger \phi)$, that breaks the power-counting renormalizability of the theory. The second and third terms in the second line of Eq.(A.1) implements off-shell in a BRST-invariant way the constraint in Eq.(A.3) via the Lagrange multiplier $X_1$. The $X_2$-dependent term simplifies diagonalization of the quadratic part via the transformation in Eq. (2).

$X_1$- and $\sigma'$- propagators have a relative minus sign responsible for their mutual cancellation inside loops, see Eq.(3), that holds true to all order by virtue of the constraint U(1) BRST symmetry

$$
\mathcal{S}X_1 = vc, \qquad \mathcal{S}c = 0, \qquad \mathcal{S}\bar{c} = \frac{1}{v} \left( \phi^\dagger \phi - \frac{v^2}{2} - vX_2 \right), \tag{A.4}
$$

all other fields and external sources being invariant under $\mathcal{S}$ and $c, \bar{c}$ being the constraint U(1) ghost and antighost fields.

The third line implements the usual $R_\xi$-gauge in a BRST-invariant way, $\bar{\omega}, \omega$ being the antighost and ghost fields associated with the gauge group U(1) and $b$ the Nakanishi-Lautrup field. The U(1) BRST symmetry is defined as usual according to

$$
sA_\mu = \partial_\mu \omega, \quad s\phi = ie\omega\phi, \quad s\sigma = -e\omega\chi, \quad s\chi = e\omega(\sigma + v), \quad s\bar{\omega} = b, \quad sb = 0, \tag{A.5}
$$

all other fields being invariant. In particular $X_2$ is BRST-invariant. The cohomological BRST analysis of the physical spectrum of the model is given in [12]. It turns out that the physical modes are the three transverse components of the massive gauge field $A_\mu$ and one physical scalar with tree-level mass $M$.

Finally the last line of Eq.(A.1) contains the external sources required to renormalize the theory. Being coupled to the BRST variation respectively of $\bar{c}, \sigma$ and $\chi$, they are the anti-fields [16] of the BRST differentials $\mathcal{S}$ and $s$. Invariance of the classical vertex functional under $\mathcal{S}$ and $s$ is translated at the quantum level into the Slavnov-Taylor (ST) identities in Eqs.(B.1) and (B.5).

# B  Functional identities

The functional identities controlling the theory are listed below:

- The ST identity for the constraint BRST symmetry is

$$\mathcal{S}_c(\Gamma) \equiv \int d^4x \left[ vc\frac{\delta\Gamma}{\delta X_1} + \frac{\delta\Gamma}{\delta \bar{c}^*}\frac{\delta\Gamma}{\delta \bar{c}} \right] = \int d^4x \left[ vc\frac{\delta\Gamma}{\delta X_1} - (\Box + m^2)c\frac{\delta\Gamma}{\delta \bar{c}^*} \right] = 0, \quad \text{(B.1)}$$

where in the latter equality we have used the fact that both the ghost $c$ and the antighost $\bar{c}$ are free:

$$\frac{\delta\Gamma}{\delta \bar{c}} = -(\Box + m^2)c, \qquad \frac{\delta\Gamma}{\delta c} = (\Box + m^2)\bar{c}. \qquad \text{(B.2)}$$

- The $X_1$-equation of motion, that follows from Eq.(B.1) by using the fact that the ghost $c$ is free:

$$\frac{\delta\Gamma}{\delta X_1} = \frac{1}{v}(\Box + m^2)\frac{\delta\Gamma}{\delta \bar{c}^*}. \qquad \text{(B.3)}$$

- The $X_2$-equation of motion:

$$\frac{\delta\Gamma}{\delta X_2} = \frac{1}{v}(\Box + m^2)\frac{\delta\Gamma}{\delta \bar{c}^*} - (\Box + m^2)X_1 - ((1+z)\Box + M^2)X_2 - v\bar{c}^*. \qquad \text{(B.4)}$$

Notice that the $z$-term is the only one that affects the right-hand side of the above equation in a linear way (so that no new external source is required to control its renormalization) and that contains at most two derivatives (in order to avoid inconsistencies of higher derivative theories due to the appearance of negative norm states in the physical spectrum).

- The ST identity associated to the gauge group BRST symmetry

$$\mathcal{S}(\Gamma) = \int d^4x \left[ \partial_\mu \omega \frac{\delta\Gamma}{\delta A_\mu} + \frac{\delta\Gamma}{\delta \sigma^*}\frac{\delta\Gamma}{\delta \sigma} + \frac{\delta\Gamma}{\delta \chi^*}\frac{\delta\Gamma}{\delta \chi} + b\frac{\delta\Gamma}{\delta \bar{\omega}} \right] = 0. \qquad \text{(B.5)}$$

- The $b$-equation:

$$\frac{\delta\Gamma}{\delta b} = \xi b - \partial A - \xi e v \chi. \qquad \text{(B.6)}$$

- The antighost equation:

$$\frac{\delta\Gamma}{\delta \bar{\omega}} = \Box \omega + \xi e v \frac{\delta\Gamma}{\delta \chi^*}. \qquad \text{(B.7)}$$

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
