# Peer review of "Renormalizable Extension of the Abelian Higgs-Kibble Model with a Dim.6 Derivative Operator"

_SciPost Physics Proceedings, doi:SciPost Phys. Proc. 14, 019 (2023)_

## Round 1 · Referee Report · Anonymous (Referee 1) · 2023-2-21

Report

The paper arXiv:2212.02453 discusses the renormalisation of a U(1) gauge theory, broken spontaneously by a charged scalar field, in the presence of a dimension-six, derivative operator. The manuscript corresponds to a talk presented in a conference, while more details on the same subject were published in a previous paper by the same Authors, arXiv:2206.00894, appeared in PRD.

The Authors argue that the model can be renormalised in a predictive way (by a finite number of physical parameters), thanks to (i) a formalism based on a choice of gauge-invariant field coordinates, and (ii) a specific, unique choice of the deformation (i.e. one particular dimension-six operator). It is then shown that all 1-PI correlators (involving all possible insertions of the deformation) can be renormalised in one shot by a finite set of renormalisation conditions.

While the computation appears to be careful and illustrates some non-trivial recursive relations among the correlators, I find the physical interpretation of the result somewhat misleading. Firstly, the Authors claim in the Introduction that effective theories are not predictive. However, it is well-known that effective theories remain predictive at low energies, as a finite number of operators of dimension less or equal than D exists, and thus a finite number of counterterms is sufficient to subtract all UV divergences up to dimension D.

Secondly, in an effective theory, if one fixes the values of the Wilson coefficients up to dimension D (by measurements), it is possible to make (finite) predictions for as many observables as desired. When the Authors assume a single dimension-6 operator, this is equivalent to assume that all others are measured to be zero (including those of higher dimensions). Under such strong assumption, it is not surprising that all correlators can be predicted as a function of that single Wilson coefficient. Of course, additional measurements may disagree with the assumption that all other operators were vanishing.

Therefore, there seems to be nothing special in the derivative operator selected by the Authors. The heavy formalism elaborated by the Authors is suitable to explicitly renormalise all amplitudes involving that particular operator. But the analogous result for any set of operators holds, and it could be proved with the standard renormalisation formalism for effective field theories. Thus, I suggest the Authors to reconsider and to specify the scope of their analysis before publication.

---

## Round 2 · Author Response

We thank the referee for his/her comments. According to the referee's report, we have inserted in the paper some remarks in order to clarify that the proposed construction applies both to the case when the model at $z=0$ is power-counting renormalizable and to the case when the model at $z=0$ is an effective field theory.

---

## Round 2 · List of Changes

The following changes have been made:
- at page 2 the third paragraph has been added in order to clarify predictivity of the effective field theories up to a given energy scale $\Lambda$
- at page 3 the last four paragraphs of Sect. 1 have been expanded in order to state that the main result of the paper, namely the algebraic reconstruction of the amplitudes of the model at $z\neq 0$ via the solution to the $z$-differential equation, also applies to effective field theories
- the second and third paragraphs of the Conclusion have been correspondingly reformulated.

---

## Editorial Decision

published